# Understanding Cell Model Characteristics—RNA Expression Profiling in Primary and Immortalized Human Mesothelial Cells, and in Human Vein and Microvascular Endothelial Cells

**DOI:** 10.3390/cells11193133

**Published:** 2022-10-05

**Authors:** Iva Marinovic, Maria Bartosova, Rebecca Herzog, Juan Manuel Sacnun, Conghui Zhang, Robin Hoogenboom, Markus Unterwurzacher, Thilo Hackert, Aurelio A. Teleman, Klaus Kratochwill, Claus Peter Schmitt

**Affiliations:** 1Center for Pediatric and Adolescent Medicine, University of Heidelberg, 69120 Heidelberg, Germany; 2Christian Doppler Laboratory for Molecular Stress Research in Peritoneal Dialysis, Department of Pediatrics and Adolescent Medicine, Medical University of Vienna, 1090 Vienna, Austria; 3Division of Pediatric Nephrology and Gastroenterology, Department of Pediatrics and Adolescent Medicine, Comprehensive Center for Pediatrics, Medical University of Vienna, 1090 Vienna, Austria; 4Zytoprotec GmbH, 1090 Vienna, Austria; 5General, Visceral and Transplantation Surgery, Heidelberg University, 69120 Heidelberg, Germany; 6German Cancer Research Center (DKFZ), 69120 Heidelberg, Germany

**Keywords:** mesothelium, endothelium, RNA sequencing, in vitro, cell models

## Abstract

In vitro studies are essential in pre-clinical research. While choice of cell lines is often driven by handling and cost-effectiveness, in-depth knowledge on specific characteristics is scant. Mesothelial cells, which interact with endothelial cells, are widely used in research, including cancer and drug development, but have not been comprehensively profiled. We therefore performed RNA sequencing of polarized, primary peritoneal (HPMC) and immortalized pleural mesothelial cells (MeT-5A), and compared them to endothelial cells from umbilical vein (HUVEC) and cardiac capillaries (HCMEC). Seventy-seven per cent of 12,760 genes were shared between the 4 cell lines, 1003 were mesothelial and 969 were endothelial cell specific. The transcripts reflected major differences between HPMC and MeT-5A in DNA-related processes, extracellular matrix, migration, proliferation, adhesion, transport, growth factor- and immune response, and between HUVEC and HCMEC in DNA replication, extracellular matrix and adhesion organization. Highly variable shared genes were related to six clusters, cell tissue origin and immortalization, but also cell migration capacity, cell adhesion, regulation of angiogenesis and response to hypoxia. Distinct, cell type specific biological processes were further described by cellular component-, molecular function- and Reactome pathway analyses. We provide crucial information on specific features of the most frequently used mesothelial and endothelial cell lines, essential for appropriate use.

## 1. Introduction

In vitro studies using specific cell models are an integral part of biomedical research, allowing for high throughput analysis of complex cellular systems. They are required for understanding (patho)physiology and for identifying interventions to then be validated experimental in-vivo studies and subsequent clinical trials. The choice of the in vitro cell model depends on the scientific questions, but also on the availability, handling and costs, but should primarily be driven by the suitability of the cells to provide a valid answer. Therefore, in-depth knowledge of specific cellular features is necessary.

Mesothelial cell (MC) monolayers line the peritoneal, pleural and pericardial cavities and the reproductive organs. They secrete lubricants to facilitate friction-free organ movement and play a critical role in local homeostasis and immune response, secreting inflammatory mediators, growth factors, extracellular matrix components and procoagulant agents [1]. MCs control fluid and solute transport [2] and they undergo mesothelial-to-mesenchymal transition (MMT) to mediate physiological tissue repair and, in case of pathological settings, angiogenesis and fibrosis [3]. Immortalized pleural mesothelial cells (MeT-5A) and human primary peritoneal mesothelial cells (HPMC) are the most frequently used in vitro models of mesothelial cells, with the latter being suitable for a few passages only and requiring repeated, standardized isolation from human tissue. Extended studies including mesothelial gene KO models are usually performed in MeT-5A [4]. MCs are attached to the basal membrane, and the subjacent submesothelial connective tissue, providing blood supply via capillaries and nerves [5].

Endothelial cells (EC) form the inner lining of the vascular system. They are metabolically highly active, play a critical role in the flow of solutes and fluids into and out of the surrounding tissue [6], and control hemodynamics, coagulation and immune responses. Endothelial alterations are an essential part of a plethora of diseases, such as cardiovascular disease and cancer, the two leading causes of death worldwide [7]. ECs are tissue- and vessel type-specific [8,9]. The most widely used human umbilical vein endothelial cells (HUVEC) represent the large, venous part of the vessel tree, whereas human cardiac microvascular endothelial cells (HCMEC) are representative of microvessels. Both are primary cell lines, with HCMEC being more expensive and more difficult to culture.

To overcome the lack of essential knowledge on MC and EC lines, we performed RNAseq analyses of the four most widely MC and EC cell lines, grown on Transwells for specific polarisation. MCs and ECs are both polarized cells, but when grown on plastic or glass surfaces in vitro, they do not develop polarisation, which is a prerequisite for the expression of specific proteins [10]. Our comprehensive and in-depth analysis provides essential information on the suitability of each cell line for specific research questions based on their specific gene expression profiles.

## 2. Materials and Methods

### 2.1. Cell Culture

Human endothelial umbilical vein cells (HUVEC), human cardiac microvascular endothelial cells (HCMEC) and the immortalized mesothelial cell line (MeT-5A) were purchased from established vendors (HUVEC and HCMEC from Promocell, Heidelberg, Germany; MeT-5A (ATCC^®^ CRL-9444™) from LGC Standards, Wesel, Germany). Endothelial cells were grown in endothelial cell growth medium (Promocell, Heidelberg, Germany) with supplements and antibiotics according to the manufacturer’s instructions. MeT-5A were cultured in Medium 199 (M199, 31150022, Gibco, Thermo Fisher Scientific, Waltham, MA, USA) supplemented with 10% (*v*/*v*) foetal bovine serum (FBS, Gibco, Thermo Fisher Scientific, Waltham, MA, USA), 1% (*v*/*v*) penicillin/streptomycin (P/S, Gibco, Thermo Fisher Scientific, Waltham, MA, USA). Human peritoneal mesothelial cells (HPMC) were isolated from four non-uremic patients and cultured as previously described [11], as approved by the Ethics Committee of the Medical Faculty, Heidelberg University (S-501/2018). Informed written consent was signed by the patients. The cells were grown in M199 medium supplemented with 10% FBS, 1% penicillin/streptomycin, 0.5 μg/mL insulin, 0.5 μg/mL transferrin, 0.4 μg/mL hydrocortisone and 2 mM L-glutamine (all from Merck, Darmstadt, Germany).

### 2.2. RNA Isolation and Sequencing

Cells were seeded at a density of 2 × 10^5^ cells per well on polyester mesh (24 mm Transwell^®^, 0.4 µm pore size, 6-well type; Corning, MA, USA) under normal culture conditions in 3 technical replicates per group. HCMEC and HPMC were pooled (4 donors) before seeding while HUVEC were bought pooled. The insert and outer chamber were filled with 1.5 and 3 mL of cell culture medium, respectively. Transepithelial electrical resistance (TER) was measured daily by EVOM volt/ohm meter with STX-2 electrodes (World Precision Instruments, Sarasota, FL, USA) and cell pellets were collected after reaching plateau. Total RNA isolation was performed using RLT buffer with 1% ß-mercaptoethanol and purified using Micro RNAeasy Kit with on column DNA digestion (Qiagen, Hilden, Germany) following the manufacturer’s procedure. The RNA concentration was measured using a Qubit kit (Invitrogen, Heidelberg, Germany). RNA integrity quantification was performed using RNA screen tape (Agilent, Santa Clara, CA, USA). Detected RNA integrity number (RIN) values of samples were over 9.0 and 500 ng/sample were used for sequencing. The library was prepared using the TruSeq Stranded mRNA Library Prep Kit (Illumina, San Diego, CA, USA). Paired-end 100 bp sequencing was performed on an Illumina Novaseq™ 6000 (LC-Bio Technology CO., Ltd., Hangzhou, China) following the vendor’s recommended protocol. The raw RNAseq data has been submitted to ArrayExpress under the accession number E-MTAB-12021.

### 2.3. Transcriptomics Data Processing

Data processing was performed using R (www.r-project.org, accessed on 10 October 2021). Background correction was performed using negative control probes which represented an empty line containing no cell material. The R package Rsubread (available from http://www.bioconductor.org, accessed on 11 October 2021) was used for genome alignment, mapping of reads and producing a matrix of counts [12,13,14]. Out of 60,666 ProbeIDs on the chip, 12,760 protein-coding transcripts remained after filtering out transcripts with an average raw count value < 100, or delta < 1000 and raw count value < 200 in one of the groups. For non-protein-coding transcripts raw counts > 3 were included. For comparisons of two cell models, only the two groups in focus were considered for applying the exclusion criteria. Data normalization and quantitation of differential abundances were performed using the R package DESeq2 [15]. Student’s *t*-test *p*-values were additionally corrected for multiple testing with the Benjamini-Hochberg method.

### 2.4. Gene Enrichment Analysis and Visualization of Data

Gene ontology (GO) enrichment analysis was conducted on three different aspects: biological process (BP), cellular component (CC), and molecular function (MF) using ClueGO (Cytoscape app, v.2.5.8., GO database 19 September 2021) [16] and PANTHER online database (https://doi.org/10.5281/zenodo.5228828, released: 18 August 2021). Reactome pathway enrichment analysis was performed using ClueGO. For the gene set enrichment analysis (GSEA) tool, the gene sets were ranked by log2-fold change (FC) value (HUVEC vs. HCMEC, HPMC vs. MeT-5A). The ranked gene list was imported into the GSEA software (v.4.1.0., http://www.gsea-msigdb.org/gsea/index.jsp, accessed on 28 September 2021) [17]. The gene set database “GO: Gene Ontology gene sets (c5.all.v7.5.1.)” was downloaded from MSigDB and queried for pathway enrichment analysis. The size of detected gene sets was limited to 15 to 350 genes. Resulting pathways were selected using FDR Q < 0.05. The EnrichmentMap add-on (Cytoscape) was applied to represent the significant gene sets, to collapse redundant pathways into single biological themes and to visualize the regulation network [18]. Enriched pathways were clustered according to shared genes, applying a minimum gene overlap of 37.5% for shown edges in the network.

Heatmap visualisation of DEGs was performed using the R package pheatmap [19]. Column and row clustering were performed by Ward clustering (agglomeration method “Ward.D2”) and Euclidean distance of similarity.

The clusters underwent GO enrichment analysis. All significantly enriched terms of biological processes were clustered by ClueGO based on similarity. The statistical significance threshold level for all gene ontology enrichment analysis was FDR Q < 0.05 (Benjamini and Hochberg corrected for multiple comparisons). Data were visualized using R package ggplot [20].

## 3. Results

### 3.1. Transcript Discovery Rates

We performed next-generation RNA-seq starting from total RNA from polarized cells. Out of 19,962 unique protein-coding transcripts, 12,760 passed the filtering criteria. From these transcripts, 9853 were shared between all cell lines and 366, 631, 99 and 87 transcripts were HPMC-, MeT-5A-, HUVEC- and HCMEC-specific, respectively. Shared transcripts are given in Figure 1. From 4367 non-protein-coding transcripts 1890 were non-coding RNAs and 2477 were pseudogenes. Mesothelial cell lines expressed 969 transcripts more than endothelial cell lines, with enriched biological functions comprising metabolic, cardiac, neuron, adhesion and ion transport functions (Appendix A). Mesothelial cell lines shared 393 mesothelial cell-specific genes, endothelial cell lines shared 235-cell specific genes and their enriched processes and pathways are given in Appendix A. Direct comparison of differentially expressed genes shared by mesothelial and endothelial cells yielded 174 significant genes (Appendix A), which were related to apoptosis, regulation of Notch and Rho signalling, anion and vesicle transport, response to hypoxia, connective tissue development, cell shape and response to bacteria (Appendix A). The top 50 genes most variably expressed in mesothelial vs. endothelial cells are given in a heatmap (Appendix A).

### 3.2. Comparison of Mesothelial Cell Models

HPMC and MeT-5A shared 10,624 transcripts (Figure 2A). Seven hundred and seventy-one were HPMC specific and 944 were MeT-5A specific. Transcripts unique to HPMC were related to the GO terms cell adhesion, immune response, angiogenesis, ECM organization and channel activity (Figure 2B), while transcripts unique to MeT-5A were related to catabolic processes, detection of chemical stimulus and DNA binding (Figure 2C). Of the shared transcripts, 612 were significantly more abundant and 488 less abundant in HPMC versus MeT-5A cells (Figure 2D) (FDR Q-value < 0.05, abs(log2FC) > 2). The list of all enriched cell specific gene sets is provided in Appendix A.

Differential GSEA on the shared genes (Appendix A) revealed higher abundance of 632 and lower abundance of 453 GO terms in HPMC vs. MeT-5A (FDR < 0.05). GO terms enriched amongst the highly abundant genes were immune and growth factor response, cell migration, proliferation and adhesion, ECM and junction organization, transport, ROS metabolic processes, proteoglycan binding and apoptosis (Figure 3). GO terms identified amongst transcripts less abundant in HPMC cells compared to MeT-5A cells were DNA recombination, telomere maintenance, mitosis, DNA damage response, RNA processing, ATP production and biosynthesis of bases, which reflects the transformation status of the primary versus immortalized cell lines. The largest node of GO terms amongst high abundance transcripts in HPMC related to numerous immune functions (Figure 3) and shared genes with angiogenesis, vasculature development, epithelial and smooth muscle cell proliferation and branching and cell migration, proliferation and adhesion. The term cell adhesion included focal adhesion, cell junction and actin filament organisation, and tyrosine kinase activity. GO term extracellular matrix (ECM) comprised ECM organisation, ECM structural constituent, collagen metabolism, connective tissue development and regulation of MMP activity. Transport related nodes of GO terms associated with transcripts that were more abundant in HPMC than MeT-5A involved metal ions, amino acids, fatty acids and endocytosis.

### 3.3. Comparison of Endothelial Cell Models

HUVEC and HCMEC shared 10,534 transcripts (Figure 4A). Four hundred and fifteen were HUVEC specific and 421 were HCMEC specific. Transcripts unique to HUVEC were related to parental terms embryonic development and transcription factor activity (Figure 4B), while transcripts unique to HCMEC were related to tissue and organ development, angiogenesis, mineralization, cellular response and binding processes (Figure 4C). Of the shared genes, 60 genes were significantly more abundant and 148 less abundant (Figure 4D) in HUVEC versus HCMEC (FDR Q-value < 0.05, abs(log2FC) > 2), suggesting higher similarity between endothelial cells compared to mesothelial cells. The list of all enriched cell specific gene sets is given in Appendix A.

We identified 82 differential GO terms enriched amongst the higher abundance genes and 81 GO terms amongst the lower abundance genes (Appendix A) in HUVEC vs. HCMEC (FDR < 0.05). Highly abundant GO terms were related to chromosome segregation, DNA replication and nucleosome organisation; lowly abundant GO terms were mainly related to cell adhesion, ECM organization and structure (Figure 5).

### 3.4. Cluster Analysis of Shared Transcripts between All Cell Lines

From 9853 differentially expressed genes common to all cell lines, genes with variance greater than 0.5 (*n* = 2566) were grouped into six clusters (C1–C6) based on their co-expression (Figure 6A). Column clustering revealed a higher level of similarity between HUVEC and HCMEC, which belongs to one parent cluster shared with HPMC. MeT-5A were distinct from the other three cell lines. The largest cluster, cell cycle process, was more abundant in the immortalized MeT-5A only. Circulatory system development was more abundant in endothelial cells. Regulation of vasculature development, however, was more abundant in HPMC and HCMEC, unchanged in HUVEC and suppressed in MeT-5A. The cell migration cluster was more abundant in the endothelial cell lines, unchanged in HPMC and less abundant in MeT-5A. Cell adhesion and positive regulation of angiogenesis cluster was more abundant in HPMC, the response to hypoxia was more abundant in mesothelial cells and markedly less abundant in the endothelial cell lines (Figure 6).

These findings demonstrate highly cell type specific expression profiles. The detailed terms of the individual clusters are given in Figure 6C. To further describe the functional relevance, the cellular localisation, and downstream signalling of the identified clusters, we analysed all identified clusters by cellular component (CC), molecular function (MF) and Reactome pathway analysis (Appendix A) which reflect the distinct biologic processes. 

## 4. Discussion

In vitro cell studies are an integral part of biomedical research and drug development, with the choice of in vitro models often driven by availability, handling and costs, while lacking essential knowledge on the suitability. This is especially true for MC cell lines, MeT-5A, i.e., immortalized MC derived from pleura and human primary peritoneal MC cells, all of which are currently frequently used. MCs play a key role in serosal cavity homeostasis and related diseases in oncology, cardiology, pulmonology, gynaecology, surgery and in nephrology, specifically in the context of peritoneal dialysis. We now provide the first comprehensive analysis of gene expression profiles of the two types of most frequently used MCs for in vitro research. Since mesothelial and endothelial cells interact in vitro in co-culture systems [21] and in vivo [22], we compared the two MC types with the most frequently used EC types, HUVEC, which are large human vessel endothelial cells, and HCMEC, which are human microvessel endothelial cells. All cells were grown on Transwell filters to assure cell polarization similar to physiological conditions. This approach increases the sensitivity and specificity in detection of differential gene expression related to bioprocesses such as angiogenesis, cell adhesion, migration and the extracellular matrix [23,24].

The four cell types, which are all derived from the same developmental origin, the mesoderm [23], share a large number of expressed transcripts under control conditions. MCs and ECs both represent functional barriers lining the serosal cavities and the vasculature, respectively. In our analysis, MCs express roughly 8% more genes than endothelial cells, with biological functions related to the local physiology of their respective organs and tissues, e.g., the heart, and specific functions of the serosa such as pain perception [25].

Comparing HPMC to MeT-5A, 7% of all transcripts expressed by the two MC types are cell specific and not shared between the MC cell lines. MeT-5A specific profiles are enriched for DNA related processes and mitosis, which is in line with their higher proliferation as compared to HPMC, and for catabolic processes and detection of chemical stimulus. HPMC specific transcript clusters are related to cell adhesion, immune response, angiogenesis, ECM organization and channel activity. When comparing the overlapping but differentially expressed genes, cells adhesion, immune response, angiogenesis, ECM organisation and transport related processes were most prominent. These findings demonstrate fundamental differences in suitability for specific research questions, as discussed below. The higher expression of cell cycle related genes in MeT-5A may be related to immortalisation process. The higher expression of immune related genes in HPMC may be observed due to the tissue origin, with peritoneal MC possibly being exposed more often to pathogens invading from the intestine into the peritoneal cavity than pleural MC.

As for MC, most of the transcripts expressed under control conditions were shared between HUVEC and HCMEC, but again with distinct differences in expression levels. Gene sets enriched in transcripts that were more abundant in HUVEC as compared to HCMEC involved chromosome segregation and cell division, gene sets enriched in less abundant transcripts related to extracellular matrix regulation and cell adhesion. Genes specific to HUVEC regulate embryonic development and reflect their foetal origin, while 15 HCMEC specific gene sets were identified and comprise an array of important biological functions, including angiogenesis. Synthesis and angiogenic response to angiogenic factors differs substantially between HUVEC and HCMEC [26]. Likewise, the response of HUVEC to retinoic acid also differs compared to dermal and pulmonary microvascular endothelial cells [27]. The vascular system is diverse in structure and physiology and lining endothelial cells have distinct gene expression programs, which impact their response to experimental manipulation, and their interaction with other cell types [28]. Careful endothelial cell selection is necessary when studying specific processes, as discussed below. We recently compared the proteome of human omental arterioles and HUVEC cells and found a considerable overlap in regulated proteins. Differences regarded immune system processes, hardly detectable in HUVEC and proteins associated with the extracellular region were less evident [29].

Comparison of the most variable transcripts that were still common to all four cell types in this study showed the highest degree of similarity between the two endothelial cell types, followed by primary mesothelial cells, while MeT-5A were the most distant. As expected, the circulatory system development represented the most specific endothelial cluster. Reactome analysis revealed the RhoGTPases as most enriched, but were comparable between both types of endothelial cells. The second cluster, including genes involved in cell migration processes and vesicular transport, was heavily downregulated in MeT-5A cells, while being preserved in the three cell lines. The cluster summarized as cell cycle process, was associated with transcripts with higher abundance in the MeT-5A. These findings suggest major functional differences, possibly related to the immortalisation status. MeT-5A were immortalized with SV40 early region genes, which have been shown to induce the expression of cytoprotective calretinin [30] in mesothelial cells and to disrupt the actin cytoskeleton and tight junctions in kidney epithelial cells [31].

The cluster regulation of vasculature development and response to growth factor stimulus was pronounced in HPMC and HCMEC, less so in HUVEC and completely suppressed in MeT-5A. Mesothelial cells impact angiogenesis in peritoneal tissues via secretion of proangiogenic factors such as VEGF, angiopoietin and chemokines such as CXCL1 [22], and even more so after transition to a mesenchymal cell type [3]. This is relevant in the context of PD, with peritoneal vessel density predicting peritoneal membrane transport function [32], but also in the context of peritoneal carcinomatosis [33]. In view of the present findings, in co-culture experiments for in-depth understanding of cellular interactions of MC and EC, HPMC or MeT-5A should be carefully selected depending on the readout of the experiments.

Mesothelial cells are exposed to major mechanical shear stress, and thus, tight adhesion to the peritoneal basement membrane is needed to prevent the exceeding loss of the mesothelial cell lining. Consistent with this, HPMC showed, by far, the strongest expression of genes involved in cell adhesion, while MeT-5A were similar to HUVEC and HCMEC. These venous and capillary endothelial cells are exposed to low blood pressure and thus less shear stress, as for example, aortic endothelial cells. Use of MeT-5A in experimental studies in this context again should be critically assessed and obtained results should be evaluated in HPMC. The last cluster identified in the comparison of the four cell lines was the response to hypoxia, with similar gene expression levels in both MC lines. This is in line with a previous study, demonstrating a similar HIF-1α mediated MMT response of rat primary peritoneal MC and MeT-5A in response to hypoxia [34]. The significantly lower expression levels of genes involved in the response to hypoxia in HUVEC and HCMEC may reflect the differences in physiological oxygen supply, which is persistently high for the endothelial lining of the umbilical vein and microvessels. Mesothelial cells, however, cover basement membranes of the intestine and the lung, with oxygen supply achieved by diffusion only, and possibly fluctuating with mechanical stress [35].

In addition to the cluster analysis providing a categorization of overarching differences between the four cell lines, we further described the specific biological processes by cellular component-, molecular function- and Reactome pathway analyses, allowing for detailed search for processes of interest when designing experimental in vitro studies. Noteworthy, 771 and 944 genes were only expressed in HPMC and MeT-5A, respectively. A total of 415 genes were HUVEC and 421 genes HCMEC-specific. This represents essential information when specific components of biological processes and gene regulation studies are envisaged.

The commercially available HUVEC are from pooled donors. For adequate comparisons we therefore pooled HCMEC and HPMC from four individual donors, introducing donor-specific variations. Despite this, we obtained highly consistent findings when comparing the cell lines and technical variability was excellent. We previously described the transcriptome of HPMC from four different donors [36], the heterogeneity between different donors was low. We cultured all cell lines in Transwell systems to obtain cell polarization. While this increased the sensitivity and specificity of our comparative analyses, we cannot rule out that different findings may have been obtained when comparing cells cultured on conventional plates. The latter, however, should rather decrease the validity of the cell model studies as compared to the in vivo setting, where endothelial and mesothelial cells adhere to a basement membrane on the basolateral site and interact with the peritoneal and vascular lumen on the apical site.

## 5. Conclusions

Our in-depth RNAseq analysis of the MC and EC types most frequently used in in vitro studies provides a plethora of essential information on cell specific features, necessary for appropriate use in experimental single cell line and cell co-culture studies.

## Figures and Tables

**Figure 1 cells-11-03133-f001:**
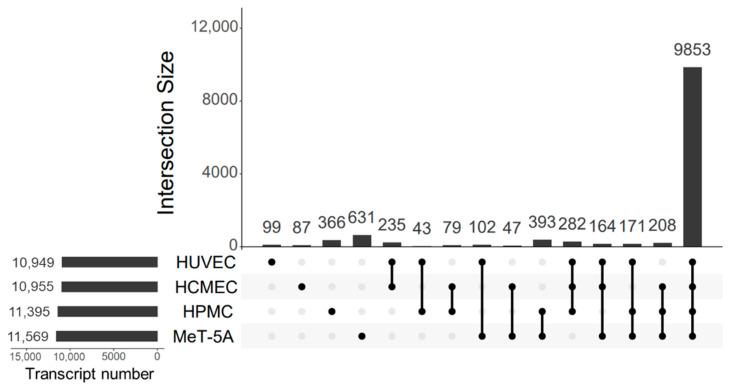
Identified protein-coding transcripts and their presence in studied cell lines. From a total of 12,760, 9853 transcripts were shared between all cell lines, 366, 631, 99 and 87 transcripts were HPMC-, MeT-5A-, HUVEC- and 87 HCMEC-specific, respectively.

**Figure 2 cells-11-03133-f002:**
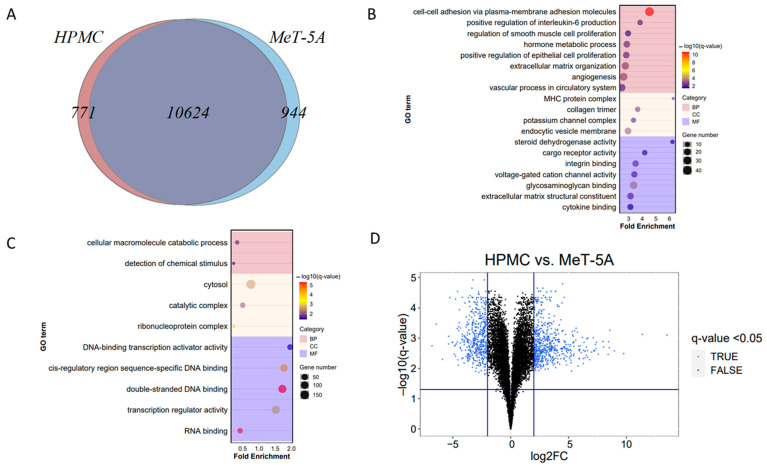
Mesothelial cell model comparisons (**A**) Venn diagram of differentially expressed genes in human primary peritoneal mesothelial cells (HPMC) versus immortalized pleural mesothelial cell line (MeT-5A). (**B**) Gene Ontology (GO) term enrichment analysis of 771 HPMC-specific genes. Number of genes per GO term and fold enrichment are given by bubble size and on the *x*-axis, respectively. The pink categories represent significantly regulated biological processes (BP), the light-yellow categories cellular components (CC), the purple categories blue molecular function (MF), as identified by Panther Gene Ontology Enrichment Analysis (FDR Q-value < 0.05). (**C**) 944 MeT-5A-specific genes (**D**) Volcano plot of differentially regulated transcripts in HPMC vs. MeT-5A (FDR Q-value < 0.05, abs(log2FC) > 2), 488 genes were less and 612 were more abundant in HPMC.

**Figure 3 cells-11-03133-f003:**
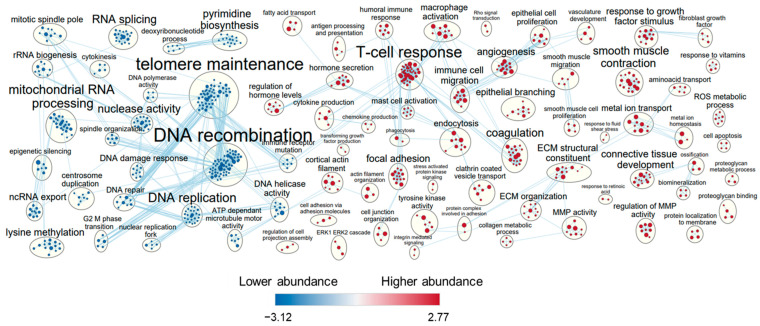
Gene set enrichment analysis (GSEA) of differentially expressed genes (DEGs) in mesothelial cells. Of the 10,624 genes shared by HPMC and MeT-5A, significantly different regulation (FDR Q-value < 0.05) of gene sets as identified by GSEA are shown. Nodes of GO terms of more abundant gene sets are represented in red, and of less abundant gene sets in blue. Node size reflects the gene set size and letter size indicates the size of the summary term. Genes shared by gene sets are reflected by connecting lines.

**Figure 4 cells-11-03133-f004:**
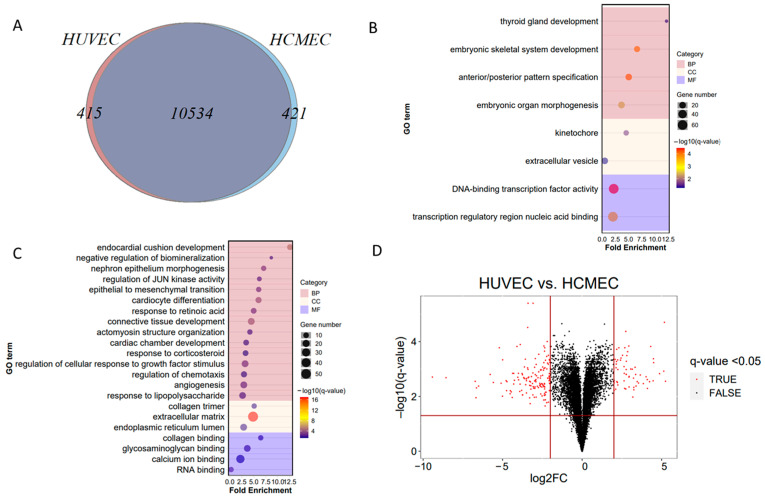
Endothelial cell model comparison. (**A**) Venn diagram of differentially expressed genes in Human Umbilical Vein Endothelial Cells (HUVEC) versus Human Cardiac Microvascular Endothelial Cells (HCMEC) (**B**) Gene Ontology (GO) term enrichment analysis of 415 HUVEC-specific genes. Number of genes per GO term and fold enrichment are given by bubble size and on the *x*-axis, respectively; the pink categories represents significantly regulated biological processes (BP), the light-yellow categories cellular components (CC), the purple categories blue molecular function (MF), as identified by Panther Gene Ontology Enrichment Analysis (FDR Q-value < 0.05) (**C**) 421 HCMEC-specific genes (**D**) Volcano plot of differentially regulated transcripts in HUVEC vs. HCMEC (FDR Q-value < 0.05, abs(log2FC) > 2), 148 genes were less and 60 were more abundant in HUVEC.

**Figure 5 cells-11-03133-f005:**
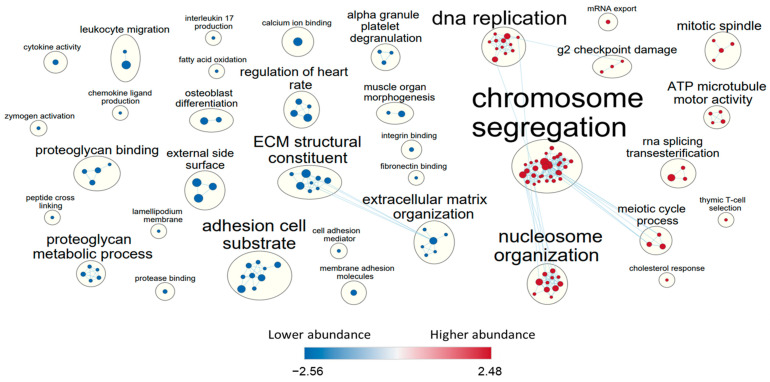
Gene set enrichment analysis of differentially expressed genes (DEGs) in endothelial cell models. Of the 10,534 genes shared by HUVEC and HCMEC, significantly different regulation (FDR Q-value < 0.05) of gene sets as identified by GSEA are shown. Nodes of GO terms of more abundant gene sets are represented in red, and of less abundant gene sets in blue. Node size reflects the gene set size and letter size indicates the size of the summary term. Genes shared by gene sets are reflected by connecting lines.

**Figure 6 cells-11-03133-f006:**
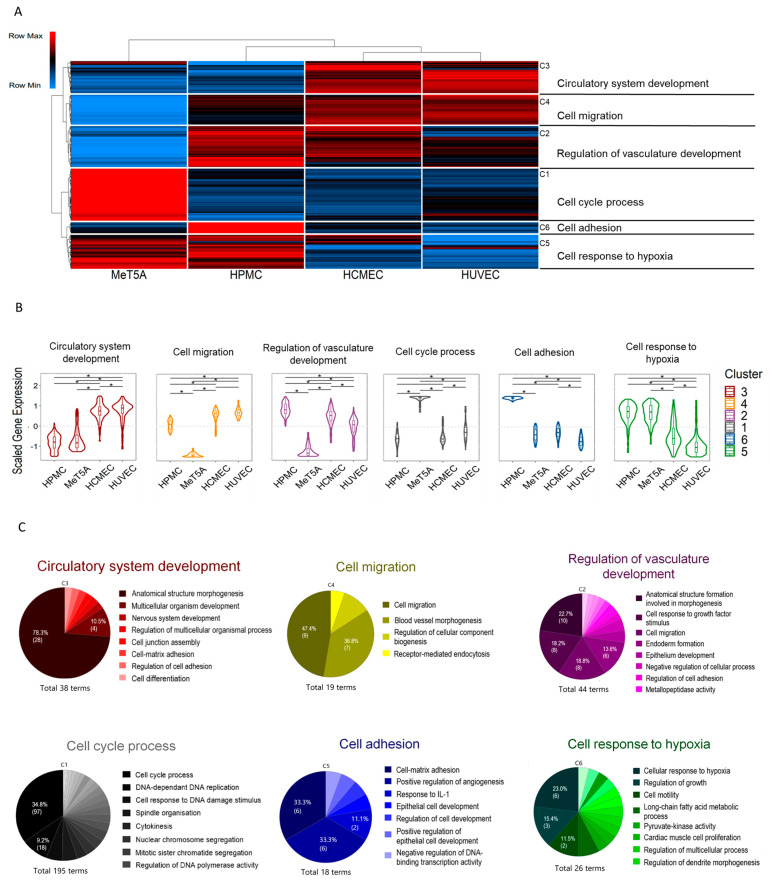
Transcriptional profiles and biological processes identification of differentially regulated clusters in the 4 studied cell lines. (**A**) Heatmap represents the expression profiles of 2566 genes with variance > 0.5. Gene Ontology (GO) enrichment analysis was used to identify biological processes represented by genes within each cluster. Representative GO terms are given. (**B**) Relative summary (values −1 to 1) of gene expression per cell line within a given cluster. Cluster numbers have been ranked in descending order to inversely correlate with the size of the term. (**C**) Cluster-specific transcriptome profiles were explored with GO enrichment analysis. All significantly enriched terms per biological process were clustered based on similarity using the ClueGO tool. Percentages and numbers of terms of the most prominent similarity-based clusters are displayed.

## Data Availability

RNAseq raw data were uploaded ArrayExpress with the accession number E-MTAB-12021. Other study related data will be made available upon reasonable request to the corresponding author.

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
