# Peer review of "Understanding Cell Model Characteristics—RNA Expression Profiling in Primary and Immortalized Human Mesothelial Cells, and in Human Vein and Microvascular Endothelial Cells"

_cells, 2022, doi:10.3390/cells11193133_

Round 1

Reviewer 1 Report

The work of Marinovic et. al. competently analyzes and compare the gene expression profile of two human mesothelial and two endothelial cell lines widely used in in-vitro studies. The work presented in this manuscript is novel in the peritoneal field, well done and has important implication for the in-vitro research and the translational ability of the derived results. The figures are of good quality, properly described and each appear necessary to document the work done.

Specific Comments:

The present manuscript investigates and analyzes, by  RNA sequencing, the expression profile of four polarized peritoneal cell types.

This manuscript is novel and original since it is the first type of study evaluating cells present in the peritoneum. It is relevant in the field since it significantly helps scientists in the choice of the most suitable in a vitro cell models.

The data presented are new and provide relevant information on cell-specific features, mandatory for appropriate use in experimental individual and mesothelial/endothelial co-culture studies.

The methods used are appropriate and correctly described.

The discussion section and conclusions are properly supported by the presented data, addressing in the meantime the main questions.

References are appropriate and reflect the actual state of the art.

The figures are of good quality, properly described and each appears necessary to document the work done.

Author Response

We thank the reviewer for his positive feedback.

Reviewer 2 Report

In this study the authors performed RNAseq on two mesothelial cell lines and two endothelial cell lines and identified differential gene expression and pathway activations between different mesothelial cells, different endothelial cells, and among all four cell lines. 

Major critiques:

1. The authors performed thorough analyses within endothelial and mesothelial groups.  Similar analysis should be done to find difference between endothelial cells and mesothelial cells.  This is more important than finding subtle differences within each cell types.

Minor critiques:

1. A heatmap should be made to show top 50 mesothelial genes and endothelial genes.

Author Response

Reviewer 2:

In this study the authors performed RNAseq on two mesothelial cell lines and two endothelial cell lines and identified differential gene expression and pathway activations between different mesothelial cells, different endothelial cells, and among all four cell lines. 

Major critiques:

  1. The authors performed thorough analyses within endothelial and mesothelial groups.  Similar analysis should be done to find difference between endothelial cells and mesothelial cells.  This is more important than finding subtle differences within each cell types.

------- We thank the reviewer for his positive evaluation. As suggested we have now amended analyses between endothelial and mesothelial cells. Figure 1 provides an upset blot comparing shared genes and cell type specific genes between all 4 cell lines, i.e. also between the endothelial and mesothelial cells (4 comparisons). Figure 6 compares transcriptional profiles and biological processes of differentially regulated clusters in the 4 cell lines. Detailed information on each of the 6 clusters is given in Suppl. Figures 3 to 8 (of which the legends have been amended, too). In addition, we now provide the enrichment of genes specific to both mesothelial cell lines and two the two endothelial cell lines, i.e. the endothelial and mesothelial cell type specific biological functions and pathways (Suppl. Figure 2A and B). Furthermore, we provide a figure with the GO analysis of the genes shared between the two mesothelial cell lines and the two endothelial cell lines (Suppl. Figure 2C and D).

Minor critiques:

  1. A heatmap should be made to show top 50 mesothelial genes and endothelial genes.

174 genes are significantly differently expressed between to two mesothelial cell lines and the two endothelial cell lines. We now add this info in the results section of the manuscript. In addition, we now list the top 50 single mesothelial genes and endothelial genes (suppl. Fig 2E). Moreover, the new Suppl. Figure 2 now provides the biological processes (and underlying gene numbers) and significance levels.

Reviewer 3 Report

Marinovic et al. examined gene expression profiles by RNAseq between cultured primary peritoneal (HPMC), MeT-5A cell line, cultured endothelial cells from the umbilical vein (HUVEC), and the cultured endothelial cells from the cardiac capillaries (HCMEC). The authors found the higher expression of cell cycle-related genes in MeT-5A and the higher expression of immune-related genes in HPMC. These cultured cells act as surrogates to evaluate biological phenotypes in vivo, meaning that comparing gene expression profiles in cultured cells to those in vivo is critical. Moreover, the authors pooled HCMEC and HPMC cells from four donors and used the pooled HUVEC cells. This pooling makes things challenging to evaluate the data since these endothelial cells have heterogeneity, and the heterogeneity should be individually different. Also, the composition of pooled cells is unclear. Thus, the reviewer feels challenging to find an advantage and novelty in the present manuscript. It may be beneficial to examine which points are comparable between in vivo and in vitro and which are not.

Minor point:

1. In line 128, please explain what negative control probes are.

Author Response

Reviewer 3

Marinovic et al. examined gene expression profiles by RNAseq between cultured primary peritoneal (HPMC), MeT-5A cell line, cultured endothelial cells from the umbilical vein (HUVEC), and the cultured endothelial cells from the cardiac capillaries (HCMEC). The authors found the higher expression of cell cycle-related genes in MeT-5A and the higher expression of immune-related genes in HPMC. These cultured cells act as surrogates to evaluate biological phenotypes in vivo, meaning that comparing gene expression profiles in cultured cells to those in vivo is critical.

-------------- The purpose of our study is the first-time comprehensive description of the suitability of the most frequently used mesothelial and endothelial cell lines for in vitro studies. In the introduction, we carefully describe the essential role of in vitro studies in research, which for obvious ethical reasons should precede in-vivo studies. We agree with the reviewer, that comparison of our findings to the in vivo situation is also of great interest. Comparing the specific findings of the cell lines in vitro with in vivo or. ex vivo findings, i.e. of microdissected mesothelial and endothelial cell layers, is part of our ongoing laborious tissue research. It will provide a large body of new information, which, however in our view exceeds the purpose of the present study by far and deserves a separate publication. As given in the discussion we recently compared the proteome of human omental arterioles with the proteome of HUVEC and found a considerable overlap in regulated proteins. Differences regarded immune system processes, hardly detectable in HUVEC and proteins associated with the extracellular region were less evident (Herzog B et al Biomolecules 2020). This illustrates the overall applicability and the limitations of using cell lines. To the best of our knowledge there are no further such comparisons published.  

Moreover, the authors pooled HCMEC and HPMC cells from four donors and used the pooled HUVEC cells. This pooling makes things challenging to evaluate the data since these endothelial cells have heterogeneity, and the heterogeneity should be individually different. Also, the composition of pooled cells is unclear. Thus, the reviewer feels challenging to find an advantage and novelty in the present manuscript. It may be beneficial to examine which points are comparable between in vivo and in vitro and which are not.

----------------- We agree with the reviewer that pooling is a critical point. By nature, primary cells have to be derived from individual donors. We carefully selected the donors for HPMC, in order to match the process for the used commercially available endothelial cells. These cell lines are cultured under highly standardized conditions prior to analysis. We agree with the reviewer, that the individual donors introduce variability, but commercially available primary cells such as the most frequently used endothelial cells, HUVEC are usually pooled. HUVEC from single donors are much more expensive and infrequently used. To allow for equivalent levels of donor-related variability and thus adequate statistical comparisons, we pooled cells from different donors for the other cell lines too. Despite this, we obtained highly consistent findings when comparing the cell lines. Technical variability was excellent. We previously described the transcriptome of HPMC from 4 different donors (Büchel et al, PDI 2014) and concluded on the basis of our previous findings that the heterogeneity between different donors was low. We now discuss this issue in the discussion section and amended the reference.

Minor point:

  1. In line 128, please explain what negative control probes are.

 ---------- controls probe was an empty line containing no cell material used for background correction. We apologize and now amended the information in the manuscript.

Round 2

Reviewer 2 Report

The authors have addressed my critiques.  I have no further questions.

Reviewer 3 Report

All of my previous concerns are now clear.